# Radical-Scavenging Activatable and Robust Polymeric Binder Based on Poly(acrylic acid) Cross-Linked with Tannic Acid for Silicon Anode of Lithium Storage System

**DOI:** 10.3390/nano12193437

**Published:** 2022-09-30

**Authors:** Hui Gyeong Park, Mincheol Jung, Shinyoung Lee, Woo-Jin Song, Jung-Soo Lee

**Affiliations:** 1Department of Chemical Engineering, Graduate School of Chosun University, 309 Pilmun-daero, Dong-gu, Gwangju 61452, Korea; 2Department of Polymer Science and Engineering, Chungnam National University, 99 Daehak-ro, Yuseong-gu, Daejeon 34134, Korea; 3Department of Chemical Engineering and Applied Chemistry, Chungnam National University, 99 Daehak-ro, Yuseong-gu, Daejeon 34134, Korea; 4Department of Organic Materials Engineering, Chungnam National University, 99 Daehak-ro, Yuseong-gu, Daejeon 34134, Korea

**Keywords:** antioxidant, lithium-ion batteries, silicone anode, polymeric binder, tannic acid, cross-linking

## Abstract

The design of a novel binder is required for high-capacity silicon anodes, which typically undergo significant changes during charge/discharge cycling. Hence, in this study, a stable network structure was formed by combining tannic acid (TAc), which can be cross-linked, and poly(acrylic acid)(PAA) as an effective binder for a silicon (Si) anode. TAc is a phenolic compound and representative substance with antioxidant properties. Owing to the antioxidant ability of the C-PAA/TAc binder, side reactions during the cycling were suppressed during the formation of an appropriate solid–electrolyte interface layer. The results showed that the expansion of a silicon anode was suppressed compared with that of a conventional PAA binder. This study demonstrates that cross-linking and antioxidant capability facilitate binding and provides insights into the behavior of binders for silicon anodes. The Si anode with the C-PAA/TAc binder exhibited significantly improved cycle stability and higher Coulombic efficiency in comparison to the Si anode with well-established PAA binders. The C-PAA/TAc binder demonstrated a capacity of 1833 mA h g^−1^_Si_ for 100 cycles, which is higher than that of electrodes fabricated using the conventional PAA binder. Therefore, the C-PAA/TAc binder offers better electrochemical performance.

## 1. Introduction

Energy has been produced via fossil fuels for decades [1,2]. Consequently, fossil fuels have depleted significantly, leading to severe environmental issues such as global warming and greenhouse gas emissions [3,4,5,6,7]. To achieve sustainable growth globally, carbon neutralization must be ensured via carbon capture utilization and storage, renewable energy, and biomass recycling [8,9]. Among them, the re-utilization of biomass is significant, owing to its zero-carbon-dioxide (CO_2_) emissions [10]. Technological developments using biomass as energy and as products are being actively pursued [11,12].

The demand for eco-friendly and safe energy storage systems in various fields is rapidly increasing [13]. Lithium-ion batteries (LIBs) are generally used as energy storage devices in various applications, such as portable electronic devices and electric vehicles [14]. Many electrode materials have been widely investigated for use as anodes in LIBs. However, the necessity to improve the performance and energy density of LIBs has increased interest in electrode material development. Researchers have extensively investigated materials with the high lithium-ion storage capability of silicon to replace the low energy density of graphite anodes (372 mA h g^−1^) in LIBs [15]. Silicon has the highest theoretical capacity of ~3572 mA h g^−1^, which translates to the advantage of low charge/discharge potentials [16,17]. Si anodes suffer from insufficient cycle lives, originating from tremendous volume expansion, up to 300%, in addition to the severe crushing of Si particles and the excessive growth of solid–electrolyte interface (SEI) layers [14,18,19]. Consequently, Si anodes exhibit fast capacity fading, low Coulombic efficiency, and electrode degradation during cycling [20,21,22].

To overcome these problems, researchers have devised several approaches to improve the lifespan characteristics of Si anodes, such as manufacturing nano-sized Si particles, carbon coating to maintain electrical conductivity, implementing a core–shell structure to reduce persistent SEI layer formation, and developing mechanically superior binders [17,23,24,25,26,27,28,29]. Among these approaches, the development of binders has received significant attention, because it is relatively simple, and not only maintains the structural stability of the electrode, but also stabilizes the electrode–electrolyte interface [30]. The binder adheres to and holds the active materials and conductive agent close with the current collector. Although employed in only small doses, whose weight is less than 5 wt% and cost less than 2% over the entire LIB, the binder is crucial in determining the function of the battery, especially for its cycle performance. Herein, recent progress on the development of novel, eco-friendly, low-cost, and water-soluble binders which recently have gained increasing attention as a promising performance booster for LIB batteries with high energy density, are reviewed. In recent studies, several water-soluble polymer binders, such as carboxymethyl cellulose, polyvinyl alcohol, styrene–butadiene rubber, and poly(acrylic acid) (PAA), have been reported [31,32,33,34,35,36,37]. The hydroxyl (-OH), carboxyl, and amino functional groups in the polymer binder can combine effectively with the native oxide layer of the Si surface [38]. Such water-soluble polymer binders are either natural, modified, or synthesized, and they are observed with profoundly enhanced chemical/physical interactions with electrode materials, stronger mechanical adhesion, and evidently improved volume variation durability, leading to dramatic improvements in the electrochemical performance of Si-based anodes.

Tannic acid (TAc) derived from biomass is significantly cheap, abundant, and widely used in many industries, such as food, pharmaceuticals, and cosmetics [39,40]. Currently, TAc is being investigated as an eco-friendly adhesive and coating, owing to its high solubility in water, high reactivity with aldehydes, and economic feasibility [41,42]. Thus, we assume that the polyphenols of TAc may scavenge superoxide radicals during discharging/charging and improve the cycling stability of LIBs, owing to the formation of strong bonds between the binder and Si surface [41,42,43,44,45]. This not only suppresses the excessive formation of the SEI layer, but also improves the conductivity of lithium ions.

Herein, we present a cross-linkable PAA/TAc as a water-soluble binder for Si anodes. This binder inhibits side reactions during cycling because of the antioxidant characteristics of TAc, contributes to the formation of a compact SEI layer, and improves the conductivity of lithium ions [46,47,48,49,50]. The obtained cross-linked network structure of the C-PAA/TAc binder would benefit the Si anode by buffering the change in volume. Therefore, we demonstrate that a cross-linked binder can be utilized to improve the cycling performance and mitigate the large volume expansion of Si anodes upon the incorporation of lithium.

## 2. Materials and Methods

### 2.1. Materials

PAA (Across Organics, Geel, Belgium, M.W. 240,000), TAc (Sigma Aldrich, Saint Louis, MO, USA, M.W. 1701.20), methanol (MeOH) (REAGENTS, DUKSAN, Korea), 2,2-Diphenyl-1-picrylhydrazyl (DPPH) (Alfa Aesar, Ward Hill, MA, USA), MeOH (HPLC grade, DUKSAN, Korea), silicon nano-particles (50 nm, Nanostructured & Amorphous Materials, Inc., Los Alamos, NM, USA), and Super P (Wellcos, Korea) were used without further purification.

### 2.2. Methods

A 10 wt% mixed solution was prepared to synthesize cross-linkable PAA/TAc. The prepared PAA/TAc (8:2, 5:5, and 2:8) were placed in a vacuum oven at 150 °C for 2 h to conduct thermal cross-linking. They were designated as C-PAA/TAc (8:2), C-PAA/TAc (5:5), and C-PAA/TAc (2:8), respectively, and were immersed in MeOH to remove any unreacted monomers. Finally, the samples were dried overnight in a vacuum oven at room temperature.

### 2.3. DPPH Radical-Scavenging Assay

DPPH is a stable free radical that is typically used to test the radical-scavenging activity of antioxidant molecules. The DPPH radical-scavenging activities of TAc, PAA, and C-PAA/TAc were measured using ultraviolet–visible (UV-vis) spectroscopy. In terms of the radical-scavenging mechanism, the antioxidant molecule serves as a hydrogen donor, transforming the DPPH radical into its reduced form. Therefore, the radical property of DPPH is neutralized, and its color transforms from purple to yellow. Briefly, a 0.2 mM DPPH solution prepared in MeOH (HPLC grade) solution and the sample solution were mixed for 30 min at room temperature. All measurements were performed under dim-light conditions. Spectrophotometric measurements were performed at a wavelength of 517 nm.

### 2.4. Characterization

The chemical structure of C-PAA/TAc was characterized via UV-vis absorbance (Amersham Biosciences, Ultrospec 2100 pro, Uppsala, Sweden). The chemical structures of the films were analyzed using Fourier-transform infrared spectrometry (FT-IR, Nicolet6700, Thermo Scientific, Waltham, MA, USA). The thermal properties of C-PAA/TAc were investigated using differential scanning calorimetry (DSC, TA Instruments DSC25, New Castle, DE, USA) under a N_2_ atmosphere (at a temperature range of 25 °C–180 °C and a heating rate of 5 °C min^−1^) and thermogravimetric analysis (TGA, TA Instruments SDT650, New Castle, DE, USA) under a N_2_ atmosphere (at a temperature range of 25 °C–1000 °C and a heating rate of 10 °C min^−1^). The morphology of the electrodes was investigated using scanning electron microscopy (FE-SEM (MSE40) UHR FE-SEM Hitachi, Japan). To confirm the degree of chemical cross-linking, the gel content was determined using the solvent extraction method and calculated as follows:(1)$$\frac{M_{2}}{M_{1}}\;\times \;100\%=\mathrm{gel}\;\mathrm{content}$$
where M_1_ is the weight of C-PAA/TAc, and M_2_ is the weight of the extracted solvent [51]. The peel-off test was conducted using a Universal Testing Machine (Shimadzu, Japan) at an extension speed of 50 mm min^−1^. Each electrode was taped with 3M of magic tape (2.5 cm in width).

### 2.5. Preparation of Silicon Anodes for Half Cells

Silicon slurries were prepared by mixing 60 wt% of silicon nano-particles as the active material, 20 wt% of PAA and TAc solution as binders, and 20 wt% of Super P as a conducting agent in deionized water. For comparison, the binders were mixed at various mass ratios of 5:5, 8:2, and 2:8 (PAA:TAc) and labeled as C-PAA/TAc (5:5), C-PAA/TAc (8:2), and C-PAA/TAc (2:8), respectively. The mixtures were cast on Cu foil using a doctor blade, which measured 40 µm in thickness. Additionally, an electrode with a higher mass loading was fabricated by controlling the amount of deionized water in the slurries. The coated electrode was dried in a convection oven at 80 °C for 1 h, then in a vacuum oven at 150 °C for 2 h. The low (0.35 mg cm^−2^) and high (1.1 mg cm^−2^) mass loading electrodes were prepared to determine the different electrochemical performance.

### 2.6. Electrochemical Measurements

CR2032 coin-type half cells of Si anodes were assembled in a glove box (KK-011AS, KIYON, Korea) under an argon atmosphere (O_2_ < 0.01 ppm, H_2_O < 0.01 ppm). The electrodes were punched into disks with a diameter of 14 mm. A lithium foil (Wellcos, Korea) measuring 300 µm in thickness was punched into disks with a diameter of 16 mm as the counter and reference electrodes. The silicon anodes were separated using a polypropylene separator (Cellgard 2400) in a coin cell. The electrolyte was 1 M of LiPF_6_ dissolved in a mixture of ethylene carbonate (EC), and diethyl carbonate at a volumetric ratio of 3:7 with the addition of 10 wt% fluoroethylene carbonate (FEC), which was purchased from Wellcos. The galvanostatic charge/discharge performance of the pre-cycle was tested at a rate of 0.05 C, and the cycling performance of the coin cells was conducted at a rate of 0.5 C for 100 cycles in the voltage window of 0.05 to 1.0 V (vs. Li/Li^+^) at room temperature. (1 C = 3000 mA g^−1^). Electrochemical impedance spectroscopy (EIS) was performed at a 10 mV amplitude signal in the frequency range of 500 kHz to 0.1 Hz.

## 3. Results and Discussions

A schematic illustration of the experimental method is shown in Figure 1a. PAA successfully formed a cross-linked structure via a condensation reaction with TAc. Through a condensation reaction between the carboxyl group of PAA and the hydroxyl group of TAc, chain molecules with cross-linked structures were formed. The condensation reaction between the carboxyl groups of PAA and the hydroxyl group of TAc occurred at 150 °C under vacuum, which managed the ester groups via interchain cross-linking [52]. To confirm and identify the esterification reaction in the Si anode structure, FT-IR spectroscopic analysis was performed, and the test results are shown in Figure 2.

The overall structures of the TAc and the cross-linked binders are shown in Appendix A. Figure 1b shows the prepared 10 wt% mixed solution of PAA/TAc.

The carboxylic acid of PAA and the hydroxyl groups of the SiO_2_ layer on the native oxide layer of the Si surface underwent a condensation reaction to form covalent ester bonds between the binder and the nano-sized Si particles (Figure 1c). The C-PAA/TAc might have increased the number of contacts between the binders and Si particles, thus enhancing binding with the Si particles and the Cu foil.

FT-IR spectroscopic analysis in the 500–4000 cm^−^^1^ region revealed the characteristic bands for PAA, TAc, C-PAA/TAc (8:2), C-PAA/TAc (5:5), and C-PAA/TAc (2:8); the results are presented in Figure 2. The ordinary peaks at 3340 and 1028 cm^−^^1^ were assigned to the -OH stretching vibration. The peaks near 2800–3000 cm^−^^1^ and 1319 cm^−^^1^ were due to C-H stretching vibrations assigned to the -CH and -CH_2_ groups of aliphatic hydrocarbons. The characteristic peaks at 1645, 1707, and 1604 cm^−^^1^ were attributed to the C=O stretching frequency of the ester groups. The absorption band at 1448 cm^−^^1^ was attributed to the aromatic C=C of TAc. The bands at 1193 cm^−^^1^ are characteristic of C-O formed by interchain cross-linking. The high-intensity C-C stretching peak at the bands in 616–890 cm^−^^1^ were associated with the C-H of benzene rings and the O-H of alcohol vibrations. The strong interaction between PAA and TAc was the key factor affecting the stability of silicon-based electrodes. [39,53,54].

To confirm the degree of cross-linking, the gel content of C-PAA/TAc was measured for each weight ratio, as shown in Figure 3a. The gel content was the highest for C-PAA/TAc (5:5). These results indicate that cross-linking was effective when the PAA and TAc weight ratios were equal. 

Conversely, as the amount of TAc increased, cross-linking occurred less, similar to C-PAA/TAc (2:8). A similar trend was observed from the results of FT-IR, DSC, and TGA.

The thermal behavior of C-PAA/TAc was analyzed via DSC, which is a typically used method for investigating polymer composites (Figure 3b). All the samples showed similar images. C-PAA/TAc (8:2) and C-PAA/TAc (2:8) indicated glass transition temperatures (Tg) of 131 °C and 137 °C, respectively, whereas C-PAA/TAc (5:5) indicated a Tg of 158 °C. The thermal properties of C-PAA/TAc (5:5), analyzed via DSC, were significantly better than those of C-PAA/TAc (8:2) and C-PAA/TAc (2:8) [41,55].

The TGA results show that C-PAA/TAc (5:5) and C-PAA/TAc (2:8) exhibited similar thermal behaviors in the first stage (Figure 3c,d). However, they exhibited different thermal behaviors in the second stage. The first weight loss stage (˂5%) occurred from room temperature to 210 °C, which was induced by the evaporation of absorbed water from the sample. The second weight loss stage occurred from 210 °C to 320 °C. The second mass loss range of 210 to 320 °C was attributed to the decomposition of groups (e.g., oxygen-containing groups) and the oxidation of carbon. The third weight loss stage, which occurred from 320 °C to 700 °C, was caused by the decomposition of the C-PAA/TAc interaction. All samples exhibited weight losses of less than 5% below 210 °C, which is more than sufficient for Si anode applications. Hence, the good thermal stability of C-PAA/TAc (8:2), C-PAA/TAc (5:5), and C-PAA/TAc (2:8) was confirmed.

Various methods are currently used to assess the antioxidant activity of plant phenolic compounds. DPPH has been widely used to evaluate the free-radical-scavenging effectiveness of various antioxidant substances. In the DPPH assay, antioxidant activity analysis is performed based on the inhibition of the DPPH radical by antioxidants. Appendix A shows the overall structures of the antioxidants and DPPH radicals.

The optimal initial concentration of DPPH was evaluated to determine the assay sensitivity. The intensity represented in violet increased rapidly as the DPPH concentration increased from 0 to 0.2 mM (Appendix A) [56,57]. Using this method, the antiradical power of the antioxidant can be determined based on the decrease in the absorbance of DPPH at 517 nm (Figure 4a) [45,58].

Figure 4 illustrates the significant decrease in the concentration of DPPH radicals owing to the scavenging ability of TAc and a standard [58,59]. The scavenging effects of TAc and the standard on the DPPH radicals decreased in the order of C-PAA/TAc (2:8) > C-PAA/TAc (5:5) > C-PAA/TAc (8:2); quantitatively, they were 92.66%, 92.57%, and 80.50%, respectively. The DPPH free-radical-scavenging activity of TAc increased with the TAc ratio (Figure 4b) [43,60].

To confirm the electrochemical performance of the silicon-based anodes using a C-PAA/TAc binder system with weight ratios of 5:5, 8:2, and 2:8 (PAA: TAc) for LIBs, we fabricated silicon/lithium half cells. Figure 5a shows that the first galvanostatic charge/discharge voltage profiles of the Si C-PAA/TAc binder electrode were evaluated in the 0.05–1 V vs. Li/Li^+^ voltage range at room temperature (25 °C) at 0.05C. The discharge capacities of the Si C-PAA/TAc (8:2), C-PAA/TAc (5:5), C-PAA/TAc (2:8), PAA, and TAc electrodes were 2706, 2692, 2595, 2583, and 2205 mA h g^−1^_Si_ at 0.05C, respectively. The initial Coulombic efficiencies of the Si C-PAA/TAc (8:2), C-PAA/TAc (5:5), C-PAA/TAc (2:8), PAA, and TAc electrodes were 82.6%, 81.5%, 79.4%, 82.3%, and 73.3%, respectively. Consequently, the initial Coulombic efficiencies of C-PAA/TAc (2:8) and TAc were lower than those of the other electrodes because these electrodes featured weak chemical bonds, which hindered electrical conduction owing to the excessive polyphenol groups [61,62]. Meanwhile, the capacities of C-PAA/TAc (5:5) and C-PAA/TAc (8:2) were much higher than that of the Si PAA electrode, indicating that the cross-linked structure of C-PAA/TAc accommodated the volume change in Si without pulverization and contributed to the formation of a compact SEI layer [40].

Figure 5b shows the long-term cycling performances of all electrodes with different polymeric binders at 0.5 C for 100 cycles. The C-PAA/TAc (8:2), (5:5), and (2:8) electrodes retained much higher capacities after 100 cycles than the PAA and TAc electrodes. C-PAA/TAc (5:5), which indicated a higher cross-linking degree than C-PAA/TAc (8:2) and C-PAA/TAc (2:8), as confirmed in Figure 3a, exhibited a reversible capacity of 1833 mA h g^−1^_Si_ and a capacity retention of approximately 100% after 100 cycles. Meanwhile, the capacity degradation of Si PAA was observed from a high initial capacity of 1585 mA h g^−1^ to the 100th cycle of 1072 mA h g^−1^_Si_, which corresponded to a capacity retention of 67.6%.

We evaluated the adhesion strength of Si electrodes with PAA and C-PAA/TAc binders to conduct a 180° peel-off test in Appendix A. As a result, the average peel strengths of the Si electrodes with PAA, C-PAA/TAc (8:2), C-PAA/TAc (5:5), C-PAA/TAc (2:8), TAc, and PVDF were 4.08, 4.11,5.48, 2.29, 1.21, and 0.47 N, respectively. The applied Si anode with a C-PAA/TAc (5:5) binder was 5.48 N, which was higher than the Si electrodes with other binders. Furthermore, the C-PAA/TAc (5:5) binder showed that a large amount of anode materials was retained on the current collector after peeling (Appendix A). These results suggest that the increase in electrode adhesion strength is attributed to the hydrogen bonding interaction between our binder and the Si powder. Based on these results, the adhesion properties of the designed binder clearly demonstrate an advantage as a binder for Si anodes. 

In addition, the Coulombic efficiency of all electrodes was approximately 98% over the entire cycle (Figure 5c). The electrochemical performance of all electrodes is listed in Appendix A. Excellent cycle stability and high reversible capacity were achieved because TAc with numerous hydrogen bonds can interact with silicon, and the strong adhesive cross-linking between the TAc and PAA chains can hinder significant volume changes and structural collapse during continuous cycles [40,52].

To determine the resistance properties of the electrodes with different binders, we conducted EIS after 100 cycles. The resistance in the high-frequency region corresponds to the resistance of the ionic electrolyte (R_s_); the diameter of the semicircle represents the interfacial resistance of the anode material (R_SEI_); and the medium-frequency region is associated with the charge transfer resistance (R_ct_). Figure 5d shows the Nyquist plots of the Si PAA, C-PAA/TAc (8:2), C-PAA/TAc (5:5), and C-PAA/TAc (2:8) electrodes after 100 cycles. The measured resistance is presented in Appendix A. The R_s_ and R_SEI_ of all electrodes were low and indicated similar values. However, the R_ct_ of Si PAA (118.4 Ω) was much higher than those of the C-PAA/TAc (8:2), C-PAA/TAc (5:5), C-PAA/TAc (2:8), and TAc electrodes (i.e., 53.6, 55.47, 102.1, and 71.79 Ω, respectively). This explains the cross-linked network structure of the C-PAA/TAc binder, which accommodated the volume change during cycling and improved the long-term cycling capacity of Si C-PAA/TAc.

To clarify the mechanism of the C-PAA/TAc binder for restraining the volume changes in silicon during the cycles indicated in Figure 5, we conducted cross-sectional scanning electron microscopy (SEM) to understand the morphology change and to measure the thickness of the electrodes. Figure 6 shows the cross-sectional SEM images of Si PAA and Si C-PAA/TAc (5:5) before and after cycling. The thicknesses of the Si PAA and Si C-PAA/TAc electrodes before cycling were 7 and 6.5 µm, respectively ((Figure 6a,c), respectively). After cycling, the thicknesses of the Si PAA and Si C-PAA/TAc electrodes were 24 and 16 µm, respectively ((Figure 6c,d), respectively), which corresponded to expansion levels of 350% and 250%, respectively. This expansion after cycling was typically observed in the silicon anodes (Appendix A). However, the C-PAA/TAc binder demonstrated superior mechanical properties that allowed it to maintain its active materials during cycling [63].

To commercialize Si C-PAA/TAc for use in LIBs, we tested the electrodes using our stable binders for the silicon anodes at a high mass loading (~1.1 mg cm^−2^), as shown in Figure 7 and Appendix A. The discharge capacity of the C-PAA/TAc electrode with high mass loading was determined to be 2361 mA h g^−1^_Si_, and the initial Coulombic efficiency was 82.6% after the first cycle. These values were higher than those for the Si PAA electrode (1602 mA h g^−1^_Si_ and 77.0%) (Figure 7a). The electrodes were tested for over 60 cycles. The results show that the capacity retention of the Si C-PAA/TAc electrode with high mass loading was 79.8% after 60 cycles, and its discharge capacity after cycling was 1219 mA h g^−1^_Si_, which was much higher than that of the Si PAA (804 mA h g^−1^_Si_) (Figure 7b). These results indicate that the cross-linked structure prevented some of the active materials from collapsing on the current collector, despite high active material loading [64].

## 4. Conclusions

Herein, we designed an excellent cross-linked PAA/TAc binder that can be used in the silicon anodes of LIBs. We showed that TAc with abundant -OH groups effectively cross-linked with PAA to achieve high electrochemical performance. These -OH groups readily formed network structures upon cross-linking. In particular, TAc, which exhibits excellent antioxidant properties, can suppress side reactions by resident DPPH radicals during cycling through antioxidants and maintain the stability of LIBs. This not only prevents the excessive formation of an SEI layer, but also improves the conductivity of lithium ions. We presented the unique features of C-PAA/TAc as a radical-scavenging component that improves the electrochemical performance of LIBs. This study demonstrated a sustainable method for fabricating a robust polymeric binder with high electrochemical durability and excellent radical-scavenging activity. In the future, this innovative binder might provide a practical solution for silicon-based anodes.

## Figures and Tables

**Figure 1 nanomaterials-12-03437-f001:**
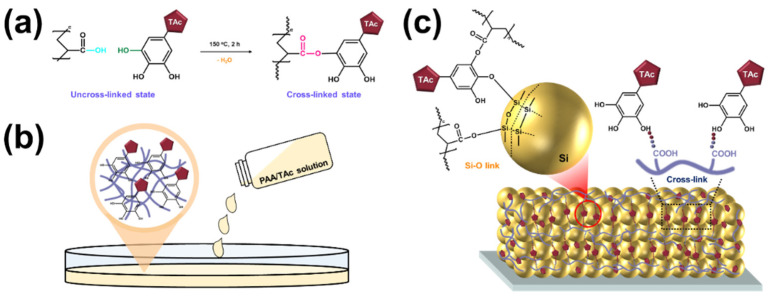
Schematic illustration showing preparation of binder using poly(acrylic acid) (PAA) and tannic acid (TAc): (**a**) PAA and TAc structure; (**b**) preparation of PAA and TAc solution; (**c**) nano-sized Si particles and interaction between C-PAA and TAc.

**Figure 2 nanomaterials-12-03437-f002:**
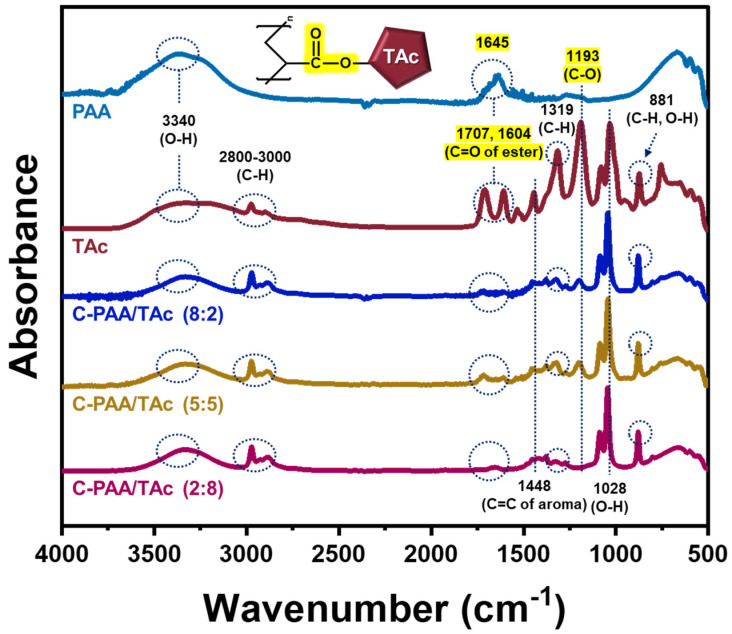
FT-IR spectra of PAA, TAc, C-PAA/TAc (8:2), C-PAA/TAc (5:5), and C-PAA/TAc (2:8).

**Figure 3 nanomaterials-12-03437-f003:**
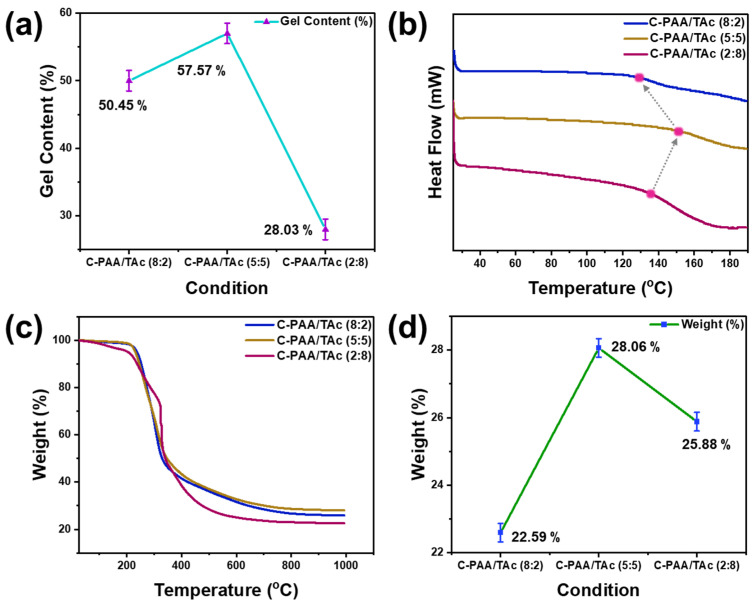
(**a**) Gel contents of C-PAA/TAc (8:2), C-PAA/TAc (5:5), and C-PAA/TAc (2:8); (**b**) DSC thermograms; (**c**) TGA thermograms; (**d**) final weight based on TGA.

**Figure 4 nanomaterials-12-03437-f004:**
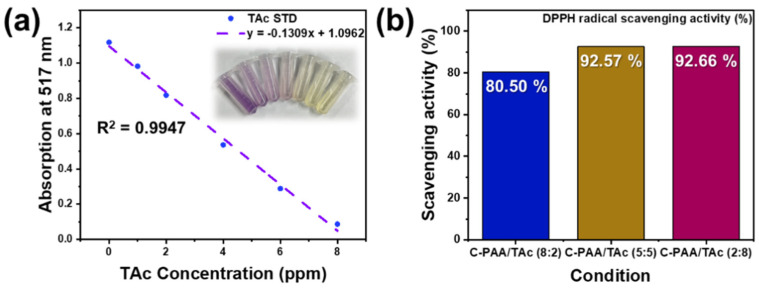
(**a**) Absorption vs. TAc concentration; (**b**) DPPH radical-scavenging activity of C-PAA/TAc.

**Figure 5 nanomaterials-12-03437-f005:**
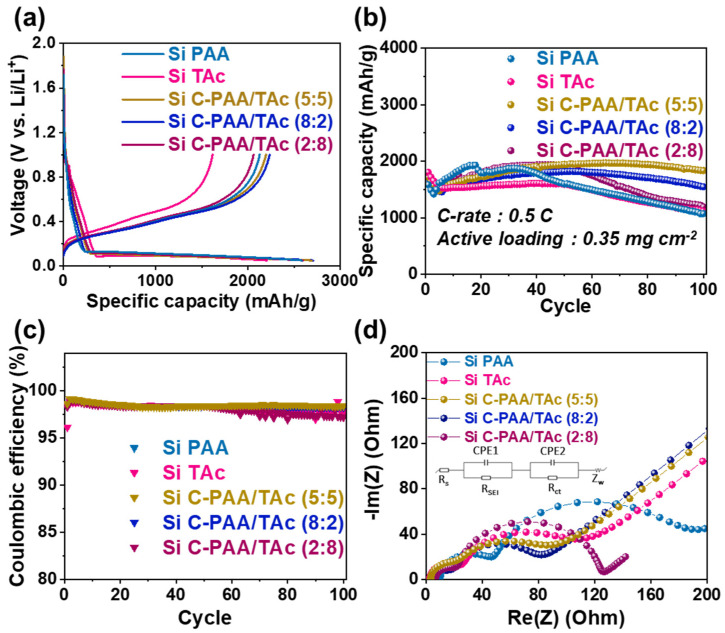
(**a**) Galvanostatic charge/discharge profiles for half cells tested by controlling binder ratio of C-PAA/TAc; (**b**) cycling performance of Si PAA, Si TAc, and Si C-PAA/TAc at 0.5 C-rate; (**c**) Coulombic efficiency of cycling performance of Si PAA Si TAc and Si C-PAA/TAc; (**d**) fitted electrochemical impedance spectroscopy of Si PAA, Si TAc, and C-PAA/TAc after 100 cycles.

**Figure 6 nanomaterials-12-03437-f006:**
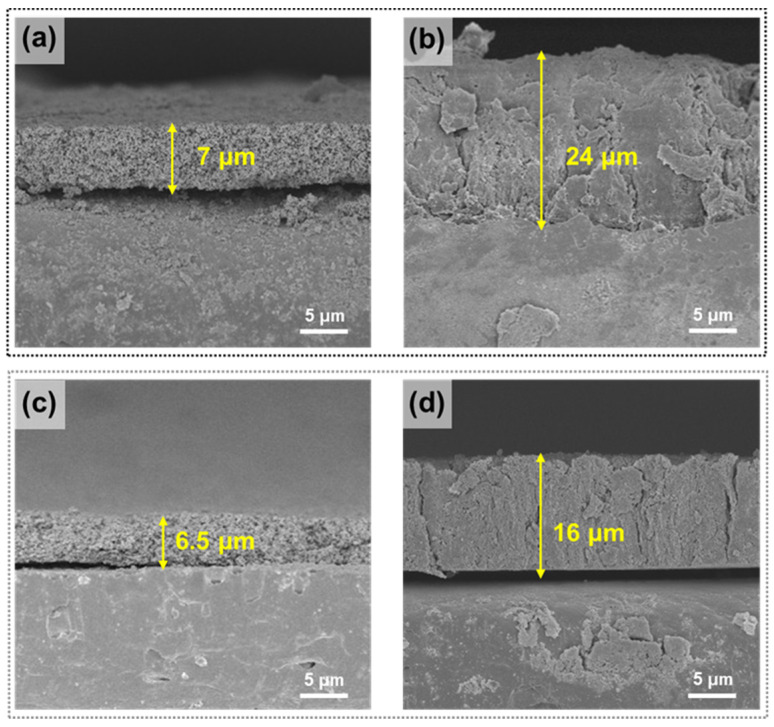
Cross-sectional SEM images: (**a**) Si PAA electrode and (**b**) after 100 cycles; (**c**) Si C-PAA/TAc (5:5) electrode and (**d**) after 100 cycles.

**Figure 7 nanomaterials-12-03437-f007:**
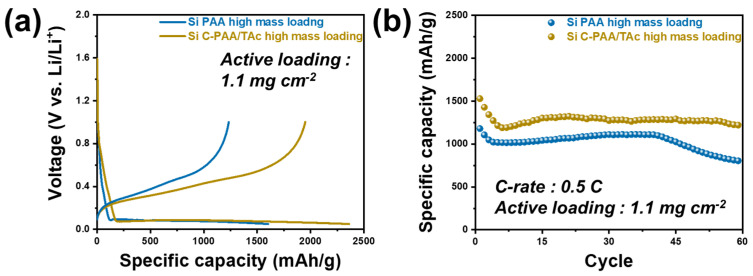
Electrochemical performance of Si ‖ Li half cells with high mass loading (~1.1 mg cm^−2^): (**a**) galvanostatic charge/discharge profiles of half cells Si PAA and Si C-PAA/TAc (5:5); (**b**) cycling performance of Si PAA and Si C-PAA/TAc at 0.5 C.

## Data Availability

Not applicable.

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
