# Peer review of "Radical-Scavenging Activatable and Robust Polymeric Binder Based on Poly(acrylic acid) Cross-Linked with Tannic Acid for Silicon Anode of Lithium Storage System"

_nanomaterials, 2022, doi:10.3390/nano12193437_

Round 1

Reviewer 1 Report

In this manuscript, the authors reported a stable network binder based on the condensation reaction between tannic acid (TAc) and poly(acrylic acid)(PAA) and applied it to the silicon anode. The C-PAA/TAc binder can improve the initial coulombic efficiency and suppress the volume expansion of the silicon anode. The authors also investigated the long-term cycling performance, resistance property and volume change of the as-prepared silicon anodes, aiming to give a full picture of this new binder for silicon anode application. Some possible mechanisms and reasons of enhanced electrochemical performance were discussed.  

1. In the first paragraph of the introduction, the authors described the background of “carbon neutral” and introduced the development prospect of biomass materials, but these contents are not relevant to the following passage. The authors should revise it or provide some useful information to enhance the connection between the full text.

2. Some expressions should be more accurate so as not to mislead the reader.
For example, in the abstract, “The C-PAA/TAc binder exhibits significantly improved cycle stability and higher Coulombic efficiency compared with well-established PAA binders”. It should be “the Si anode with the C-PAA/TAc binder exhibits significantly improved cycle stability and higher coulombic efficiency in comparison to the Si anode with well-established PAA binders” or another suitable expression.

3. In this manuscript, the authors claimed that the C-PAA/TAc can increase the contact between binders and Si particles, and further enhance the binding between Si particles and Cu foil. However, the gel content of C-PAA/TAc is not enough to support this result directly, the authors should provide the peel test results of as-prepared Si anodes.

4. The Si anode with C-PAA/TAc (5:5) has the best long-term cycling performance, but its charge transfer resistance after 100 cycles is higher than the electrode with C-PAA/TAc (8:2). Why is that?

Author Response

Reviewer #1: In this manuscript, the authors reported a stable network binder based on the condensation reaction between tannic acid (TAc) and poly(acrylic acid)(PAA) and applied it to the silicon anode. The C-PAA/TAc binder can improve the initial coulombic efficiency and suppress the volume expansion of the silicon anode. The authors also investigated the long-term cycling performance, resistance property and volume change of the as-prepared silicon anodes, aiming to give a full picture of this new binder for silicon anode application. Some possible mechanisms and reasons of enhanced electrochemical performance were discussed.

1) In the first paragraph of the introduction, the authors described the background of “carbon neutral” and introduced the development prospect of biomass materials, but these contents are not relevant to the following passage. The authors should revise it or provide some useful information to enhance the connection between the full text.

[Reply]

We changed from “Tannic acid (TAc) is significantly cheap, abundant and is widely used in many industries, such as food, pharmaceuticals, and cosmetics [38,39]” to “Tannic acid (TAc) derived from biomass is significantly cheap, abundant and is widely used in many industries, such as food, pharmaceuticals, and cosmetics [38,39].”

2) Some expressions should be more accurate so as not to mislead the reader.

For example, in the abstract, “The C-PAA/TAc binder exhibits significantly improved cycle stability and higher Coulombic efficiency compared with well-established PAA binders”. It should be “the Si anode with the C-PAA/TAc binder exhibits significantly improved cycle stability and higher coulombic efficiency in comparison to the Si anode with well-established PAA binders” or another suitable expression.

[Reply]

We thank the reviewer for his/her criticism on our work. We changed from “The C-PAA/TAc binder exhibits significantly improved cycle stability and higher Coulombic efficiency compared with well-established PAA binders” to “The Si anode with the C-PAA/TAc binder exhibits significantly improved cycle stability and higher coulombic efficiency in comparison to the Si anode with well-established PAA binders.”

3) In this manuscript, the authors claimed that the C-PAA/TAc can increase the contact between binders and Si particles, and further enhance the binding between Si particles and Cu foil. However, the gel content of C-PAA/TAc is not enough to support this result directly, the authors should provide the peel test results of as-prepared Si anodes.

[Reply]

We appreciate the reviewer for supportive comments. We evaluated the adhesion strength of Si electrode with PAA, C-PAA/TAc binder to conduct 180 o peel-off test. The anode electrodes contained silicon powder, Super P, and a binder at a weight ratio of 6:2:2. The peel tests were conducted by Universal Testing Machine (Shimadzu, Japan) at an extension speed of 50 mm min-1. Each electrode was taped with a 3M magic tape (2.5 cm in width). As a result, the peeling force applied to the Si anode with C-PAA/TAc (5:5) binder was the highest force (about 5.5 N) among the electrodes because the C-PAA/TAc (5:5) binder enhanced the binding force between Si powder and Cu foil. We included it in the revised manuscript and supporting information (Figure S5).

“We evaluated the adhesion strength of Si electrode with PAA, C-PAA/TAc binder to conduct 180 o peel-off test in Figure S5. As a result, average peel strengths of the Si electrode with PAA, C-PAA/TAc (8:2), C-PAA/TAc (5:5), C-PAA/TAc (2:8), TAc, and PVDF shows 4.08, 4.11,5.48, 2.29, 1.21, and 0.47 N, respectively, which was higher than the Si electrode with other binders. Furthermore, C-PAA/TAc (5:5) binder showed that a large amount of anode materials was retained on the current collector after peeling (Figure S5a). These results suggest that the increase in electrode adhesion strength is attributed to the hydrogen bonding interaction between our binder and Si powder. The adhesion properties of the designed binder based on these results clearly demonstrate an advantage as a binder for Si anode.”

Figure S5. (a) Optical images of the tapes peeled from the Si PVDF, Si TAc, Si C-PAA/TAc, and Si PAA electrodes; (b) 180° peel-off test results of Si PVDF, Si TAc, Si C-PAA/TAc, and Si PAA electrodes EIS data of Si PAA, Si TAc and Si C-PAA/TAc; (c) the average peeling forces.

4) The Si anode with C-PAA/TAc (5:5) has the best long-term cycling performance, but its charge transfer resistance after 100 cycles is higher than the electrode with C-PAA/TAc (8:2). Why is that?

[Reply]

The reviewer raised a very good point. To confirm more specific charge resistance, we fitted the EIS data. As a result, the difference of Rct of the Si PAA, C-PAA/TAc (5:5) was about 2Ω. We thought that this result was just an error range. We changed figure 5 (d), table 2, and the manuscript.

Figure 5(d). Fitted electrochemical impedance spectroscopy of Si PAA, Si TAc and C- PAA/TAc after 100 cycles.

Table 2. EIS data of Si PAA, Si TAc and Si C-PAA/TAc after 100 cycle.

“However, the Rct of Si PAA (118.4 Ω) was much higher than those of the C-PAA/TAc (8:2), C-PAA/TAc (5:5), C-PAA/TAc (2:8), and TAc electrodes (i.e., 53.6, 55.47, 102.1, 71.79 Ω, respectively).”

Reviewer 2 Report

The article entitled "Radical-Scavenging Activatable and Robust Polymeric Binder Based on Poly(acrylic acid) Cross-linked with Tannic Acid for Silicon Anode of Lithium Storage System" is related to a hot topic in the research field of lithium-ion batteries with silicon anodes: the development of new binders to mitigate the degradation caused by the volumetric expansion of silicon upon lithiation and avoid pulverization upon delithiation. The authors propose crosslinking of two different binders (PAA and TAc) as a strategy to enhance the binding properties and compare the influence of the fraction of each of them. The best cross-linking and electrochemical results are obtained with (5:5) ratio. The study is well organized. First, they give evidence of crosslinking and analyze the reactions occurring. Afterward, they compare the electrochemical performance of the different samples. Finally, higher loading electrodes are prepared with the most promising sample. Nevertheless, I think that the quality of the article is not high enough to be published as Nanomaterials. In my opinion, many points are open and should be clarified before being published in this or any other journal. I recommend the authors check the following comments to improve their manuscript. 

The introduction section must be improved. At some points, the authors assume that the reader knows about some issues or processes associated with the use of silicon in the negative electrode, but general readers might not know about them. Please, revise this section and make it more easily readable, better described, and coherent. In addition, I specifically propose to further discuss the literature associated with cross-linked binders for electrodes in lithium-ion batteries, as well as that of TAc. Furthermore, I believe that authors should at least mention the characteristics (pros and cons) of the state-of-the-art binders for negative electrodes in lithium-ion batteries. 

Line 49: "Silicon has the highest theoretical capacity of ~4200 mA h g -1, which translates to the advantage of low charge/discharge potentials" this statement is incorrect

Line 51: "However, Si anode materials undergo significant volume changes (> 300%) during lithiation/delithiation and exhibit low electrical conductivity [14]. This results in unsatisfactory electrical contact between conductive materials and Si particles" Re-write, please

Line 72: I think that the authors have missed something before this sentence: "Thus, we assume that the polyphenols of TAc may scavenge the superoxide radicals during the discharging/charging". No problems associated with superoxide radicals have been discussed before.

Line 79: "contributes to the formation of a compact SEI layer, and improves the conductivity of lithium ions." How does TAc improve the conductivity of Li-ions?

The authors should mention that water is used as the solvent for the anode slurry in lithium-ion batteries. The current version is not mentioned and can be difficult for inexperienced readers to understand why water-solubility (line 65) is an advantage for the binders.

Materials and methods

Please, mention the molecular weight of the binders. It is an important parameter for slurry elaboration.

Lines 87 and 88. Authors initially use "methanol (MeOH)." Thus, in the next sentence, they should type MeOH instead of methanol.

Why do authors use two different types of methanol? Please, specify in which step is used each of them. It is just described in section 2.4.

Line 99: DPPH should be introduced. What does DPPH mean?

Line 131: "the mass loading of pure silicon powders on the Cu foil was approximately 0.35 mg cm -2, and that of the electrode was about 1.1 mg cm -2." These sentences are confusing. I would recommend mentioning that low (0.35 mg/cm2) and high (1.1 mg/cm2) loading electrodes were prepared to conduct the different electrochemical experiments. 

Line 134: "Silicon anode half-cells were assembled into CR2032 coin cells (...)". Please, revise this sentence

Lines 138-140: where did the authors purchase the electrolyte from? If they elaborated it (it is not clear in the manuscript), where did they buy the different components?

Where did the authors purchase the lithium foil from?

Which potentiostats were used for the electrochemical testing?

Which were the H2O and O2 concentrations in the glovebox used to assemble the half coin cells? Which was the manufacturer of the glovebox?

After reading the full manuscript, I have observed that the authors have opened some cells after cycling. I do not see any description of the procedure followed in the experimental section. Were these cells opened in the glove box? Was any transfer chamber used?

Results

Maybe better "Results and discussion"?

Figure 2: I do not see significant differences in the intensity of the peaks associated with PAA and TAc in the crosslinked samples with different fractions of each of these compounds. Should not the peaks at 2800-3000 cm-1 be more intense for C-PAA/TAc (2:8) than for (8:2)? How authors explain this?

Figure 3c and line 201: authors describe the cause of the first and third weight losses, but omit the second one. Would it be possible to discuss it?

Line 203: "All samples exhibited weight losses of less than 5% below 210 °C, which is more than sufficient for Si anode applications." Does Si anode application have specific conditions compared with other anodes? If this was the case, please, specify them in the manuscript.

Line 221: "The scavenging effects of TAc and the standard on the DPPH radicals decreased in the order of C-PAA/TAc (2:8) > C-PAA/TAc (5:5) > C-PAA/TAc (8:2); quantitatively, they were 92.66%, 92.57%, and 80.50%, respectively." Can the authors explain why does occur such a significant drop from (5:5) to (8:2), while the value is almost the same in (2:8) and (5:5)? Which is the sensitivity of the measuring system to be able to claim that the result with (2:8) is higher than for (5:5)? 

Lines 231-237: "To confirm the electrochemical performance of the silicon-based anode using a C-PAA/TAc binder system with weight ratios of 5:5, 8:2, and 2:8 (PAA: TAc) for LIBs, we fabricated silicon/lithium half-cells comprising an electrolyte (1M LiPF 6 dissolved in EC: DEC = 3:7 (v/v) and 10 wt% of FEC as additives), as shown in Figure 5(a). The anode electrodes contained silicon powder, Super P, and a binder at a weight ratio of 6:2:2. The mass loading of the electrode was calculated based on the total weight of the active materials and was approximately 0.35 mg/cm 2." All this information has been previously provided in section 2. Please, remove it from section 3 and move Figure 5a to section 2.

Lines 239-241: I am missing some discussion about the influence of each binder on the discharge (lithiation?) capacity of the electrode. In lines 247-250, it is discussed that crosslinking is beneficial because it avoids pulverization. Nevertheless, I would expect a wider discussion, including samples with just PAA or TAc.

Are capacity values referred to as grams of Si? If this was the case, please, specify it as mAh/(g Si).

Figure 5c: why is the specific capacity initially decreased and after 5-10 cycles increased?

Lines 259-260: "In addition, the Coulombic efficiency of all electrodes showed approximately 98% over the entire cycle". I think that the word "life" is missing at the end of this sentence.

Line 267: I am not sure if "ionic electrolyte" is a concept usually used for discussion in the literature.

Line 271-274: "The R s and R SEI of all electrodes were low and indicated similar values. However, the R ct of Si PAA (144.33 Ω) was much higher than those of the C-PAA/TAc (8:2), C-PAA/TAc (5:5), and C-PAA/TAc (2:8) electrodes (i.e., 70.27, 85.10, 116.27 Ω, respectively)." this is surprising for me. I would ask if authors can find in literature examples of this phenomenon.

Line 285: "The thicknesses of the Si PAA and Si C-PAA/TAc electrodes before cycling were 8 and 6 µm" Why is the thickness of the Si C-PAA 33% thicker than that of the Si C-PAA/TAc (5:5) electrode? If this is because loading is higher as well, this could be influencing the electrochemical performance. What about the electrodes with C-PAA/TAc (2:8) and (8:2)?

Figure 6: which was the SOC of the cells from which the electrodes were analyzed?

Line 301: Why is the Coulombic efficiency higher for high-loading electrodes with C-PAA/TAc (5:5) than for low-loading electrodes?

Figure 7: why were the electrodes with high loading tested just for 60 cycles, while those with low loading were tested for 100 cycles? I would recommend comparing in a figure and comment on high loading vs. low loading.

Author Response

First, we really appreciate the valuable comments on our manuscript. According to the comments, we have thoroughly revised the manuscript, and detailed corrections are listed below point by point.

Round 2

Reviewer 2 Report

Nothing to comment.